# Study on Wind-Induced Human Comfort of the SEG Plaza under Local Excitation Based on Wind Tunnel Test

**Wei Xu** [1] **, Renjie Li** [2,*] **, Jianlei Qiu** [1,2] **, Qingxiang Li** [1] **and Zhiwei Yu** [2]

[1]  Guangdong Provincial Building Research Institute Group Co., Ltd., Guangzhou 510500, China
[2]  School of Civil Engineering, Guangzhou University, Guangzhou 510006, China
*   Correspondence: lirenjie199888@163.com

**Abstract:** Multiple unusual vibrations occurred in SEG Plaza from May 18 to 20, 2021. To investigate the causes of these vibrations, a rigidity compression wind tunnel test was applied to study the wind-induced response of the main structure, and acceleration sensitivity analysis was conducted with parameters such as wind speed, structural period and damping ratio included. Additionally, the mast vortex-induced resonance equivalent force of reaction in the bottom was exerted on the top of the structure to obtain the acceleration response of the main structure with mast. Based on the evaluation of the vibration response of the structure before and after considering the mast as per the current specifications, it is indicated that the base overturning moment of the structure is much smaller than the specification value excluding the factor of mast, and the acceleration response varies significantly with wind pressure, structural frequency and damping ratio, but the centroid acceleration and the angular acceleration meet the comfort requirements. This indicates that the wind load on the main structure is not the dominant cause of the structural vibration. With the mast taken into account, the acceleration response of the structure exceeds the limits of the comfort level to varying degrees. For a mast damping ratio of 0.3%, the maximum angular acceleration exceeds the H-90 limit and the comfort level is poor. These findings provide considerable evidence that the dominant cause of vibration in the SEG Plaza was the vortex resonance of the top mast inducing higher mode resonance in the main structure.

**Keywords:** tall building; wind tunnel test; wind-induced vibration response analysis; sensitivity analysis; wind-induced human comfort; vortex-induced vibration

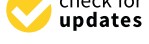



## 1. Introduction

Following the speedy development of the economy, a large number of tall/super-tall buildings have been constructed, and the research efforts of relevant technical experts on the structure of tall buildings are also gradually on the rise. Tall structures are characterized by high flexibility [1]. The vibration response of such structures under wind load is more significant, and excessive wind vibration acceleration can disrupt its normal function. On 18–20 May 2021, several unusual vibrations were observed in SEG Plaza, Shenzhen, which triggered wide social concern, and some scholars launched studies on the causes of the vibrations.

Yang Yi et al. [2] analyzed the actual wind field measurements in the planetary boundary layer of Shenzhen's atmosphere on May 18 and obtained the maximum average wind speed and wind directions during vibration. The critical wind speed for mast resonance was estimated based on the size of the top mast of the structure and that was consistent with the measured wind speed. It is therefore concluded that the vibration was caused by the specific wind condition triggering the vortex-induced resonance of the top mast of the building. Zhang Xianbing et al. [3] investigated several factors that contributed to the vibration of the building. Their analysis determined that the vibration is not related to earthquakes but suggests a possibility that internal damage to the structure led to the

vibration under wind loads. Li Junrui [4] used an autoregressive AR model to simulate wind loads and then carried out a wind-induced vibration comfort analysis on the main structure of the building, proving that wind loads on the main structure were not the main cause of vibration; rather, the vortex-induced resonance of the mast was the dominant cause of vibration. These studies focus on two factors, the vortex-induced resonance of the mast and the wind vibration simulation of the main structure, and generally conclude that the vortex-induced resonance of the mast but not the wind load of the main structure was the dominant cause of the vibrations. However, their research was undertaken under simulated wind load and other assumptions. To obtain more reasonable and solid results, a further wind tunnel test study on the main structure of the building is required and should take into account factors such as the disturbance around the building and the dynamic characteristics of the structure [5,6].

To explore the dominant causes of vibration in the SEG Plaza, rigidity compression wind tunnel testing was conducted on the main structure, the base shear and acceleration at the top of the structure under different wind directions are analyzed and sensitivity analysis of acceleration parameters is carried out with the influence of wind speed, structure period and damping ratio considered. The comfort level of the plaza is evaluated in conjunction with relevant specifications. The equivalent counter-force at the top mast vortex-induced resonance is then calculated, followed by an analysis of the acceleration response of the main structure when the factor of mast is included and, finally, an assessment of its comfort.

## 2. Overview of the Project

### 2.1. Structural Information

The project is located in Futian District, Shenzhen, at the intersection of Shennan Middle Road and Huaqiang North Road, with the real-world rendering shown in Figure 1. There are 76 floors in total in the main body, including 4 underground floors and 72 aboveground floors, with a height of 291.65 m. The 1st to 10th floors above ground are skirt buildings, as shown in Figure 2, with a total height of 50 m and a width and depth of 70.98 m. The 11th to 72nd floors above ground are tower buildings, as shown in Figure 3, with a total of 241.65 m and a characteristic width and depth of 44.4 m. The form of the structure is that of a box-and-barrel one, with the first 13 cycles and frequencies shown in Table 1, where the 2nd, 5th, 9th and 10th frequencies are the first 4 frequencies in the X direction, the 1st, 4th, 8th and 13th frequencies are the first 4 frequencies in the Y direction and the 3rd frequency is the primary torsional frequency.

**Table 1.** The first 13 cycles and frequencies.

| Vibration Pattern | Period | Frequency | Participating Mass Ratio | | |
|:---:|:---:|:---:|:---:|:---:|:---:|
| | | | UX | UY | UZ |
| 1 | 6.590 | 0.1518 | 0.04 | 0.96 | 0 |
| 2 | 6.434 | 0.1554 | 0.96 | 0.04 | 0 |
| 3 | 3.349 | 0.2986 | 0 | 0.01 | 0.99 |
| 4 | 1.715 | 0.5831 | 0 | 1 | 0 |
| 5 | 1.549 | 0.6457 | 1 | 0 | 0 |
| 6 | 1.263 | 0.7915 | 0 | 0.09 | 0.91 |
| 7 | 0.934 | 1.0710 | 0 | 0.29 | 0.71 |
| 8 | 0.819 | 1.2209 | 0.01 | 0.78 | 0.21 |
| 9 | 0.753 | 1.3286 | 0.99 | 0.01 | 0.01 |
| 10 | 0.692 | 1.4455 | 1 | 0 | 0 |
| 11 | 0.631 | 1.5848 | 0.01 | 0.11 | 0.88 |
| 12 | 0.607 | 1.6472 | 0.76 | 0.15 | 0.08 |
| 13 | 0.567 | 1.7624 | 0.12 | 0.82 | 0.06 |

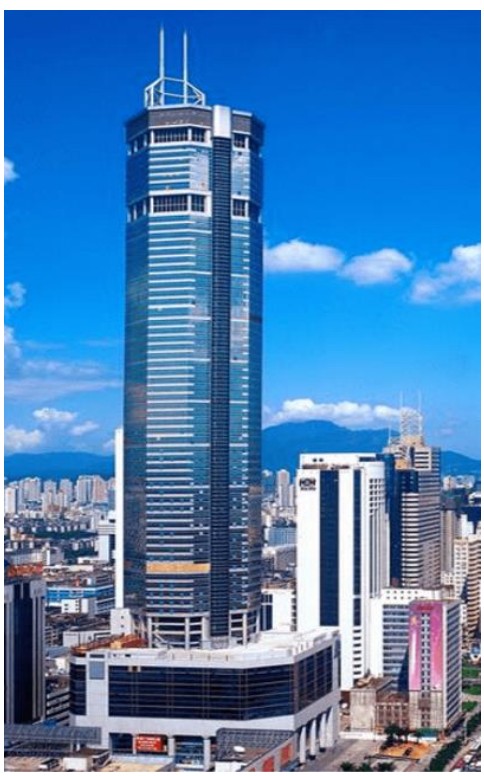

**Figure 1.** Real-life rendering.

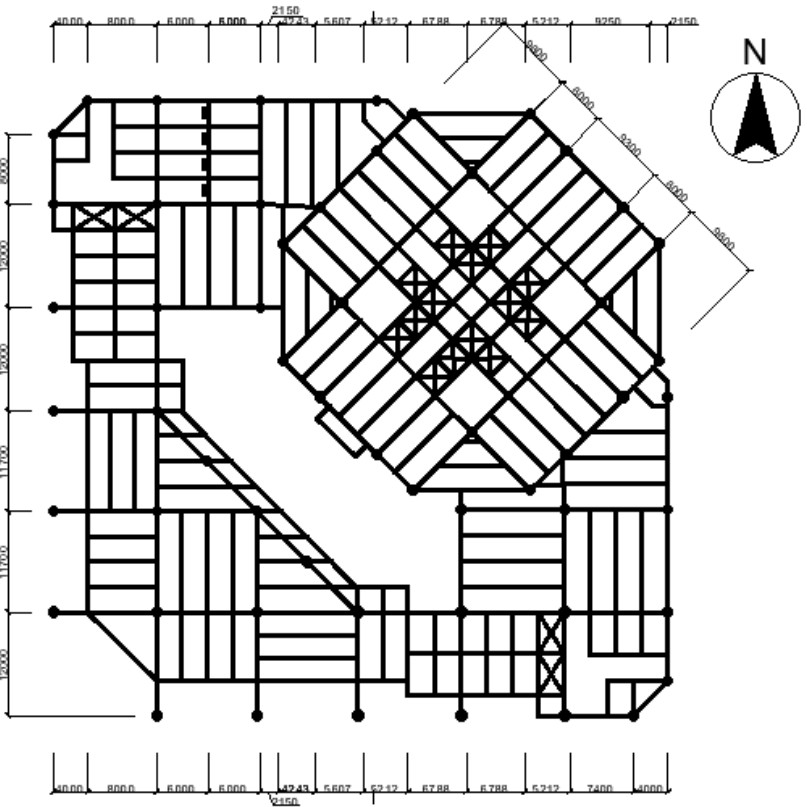

**Figure 2.** The floor plan of skirt building.

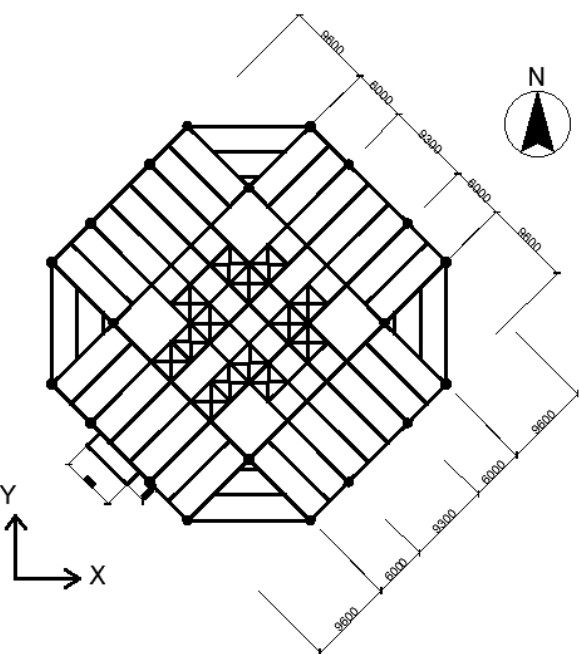

**Figure 3.** The floor plan of tower building.

### 2.2. Mast Information

The mast at the top consists of hollow round steel pipes of various sizes, as shown in Figure 4, with a total height of 54.348 m and an elevation of 291.65 m at the bottom of the mast and 345 m at the top and is divided into 11 stories, with 6 vertical rods from stories 1 to 4 totaling 28.4 m. The vertical rods are connected to each other by cross rods and diagonal rods in a lattice structure, and stories 5 to 11 totaling 25.948 m, and they are the cantilevered section of the 6 vertical rods in the lower part after the extension of the middle 2 vertical rods. The mast is located at the northeast corner of the top of the tower buildings and Figure 5 is the layout plan of the mast.

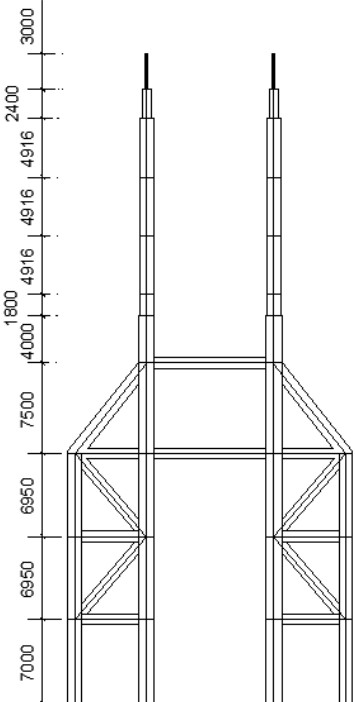

**Figure 4.** Schematic diagram of mast elevation.

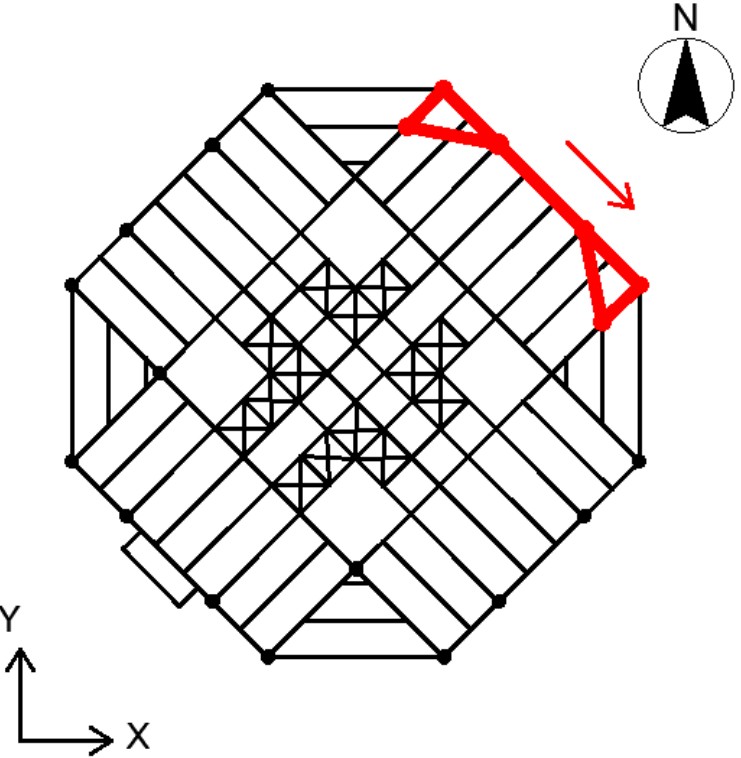

**Figure 5.** Layout plan of mast. (arrow: vibration direction of the mast).

*2.3. Real Measurements*

Hu Weihua et al. [7] collected the measured acceleration time history from 12:00 to 13:00 at noon on 20 May 2021 in the SEG Plaza, and found that four resonances occurred during the time period and the maximum acceleration at the bottom of the mast was 0.101 m/s². At around 12:54, the maximum acceleration at the bottom of the mast was 0.101 m/s², and the maximum acceleration value within the 69th floor was then 0.055 m/s². The dominant frequency of vibration at the moment of higher vibration was 2.12 Hz, at which the frequency of the 71st floor of the main body and the mast were respectively excited, and the average values of the structural damping ratio were obtained as 1.08% and 0.35%, respectively, and the maximum acceleration of 0.041 m/s² was obtained during the excitation.

Yang Yi et al. [2] analyzed the actual data of the wind field in the planetary boundary layer during the vibration. The maximum 10 min average horizontal wind speed obtained from the vibration of the building at 298 m was about 9–12 m/s, and the wind direction remained basically in the south–southwest direction throughout the day, corresponding to the 340° wind direction.

## 3. Wind Tunnel Test

*3.1. Test Overview*

The wind tunnel test is carried out on the main structure of the SEG Plaza without masts in a large scale planetary boundary layer wind tunnel with a cross-sectional size of 8 m × 5 m. The wind load is measured using the rigid model, and the rigid scale model is shown in Figure 6, with a scale ratio of 1:300 and 466 wind pressure measurement points. The test blockage ratio is 0.4%, which is less than 5%. The sampling frequency is 312.5 Hz, and each measuring point records 10,000 wind pressure data at each wind direction angle, which meets the requirements of signal sampling frequency proposed in reference [8].

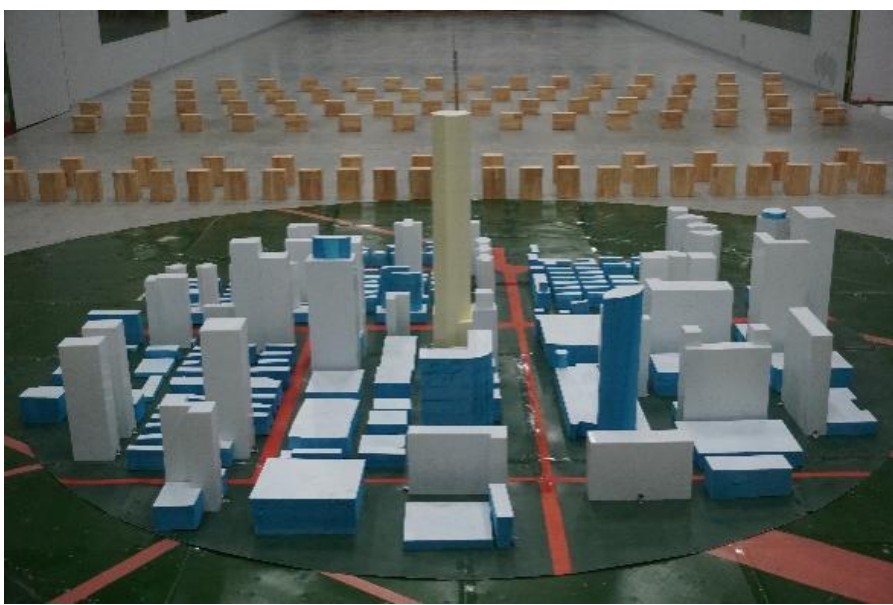

**Figure 6.** Rigid scale model.

The test was conducted in a circle at the center of the tower with the wind angle shown in Figure 7. A wind direction angle of 10° counterclockwise was established and a 500 m radius of surrounding buildings was simulated to accommodate the wind disturbance effect of surrounding buildings, together with two conditions that were set up with or without surrounding buildings. Analysis on roughness was carried out for each wind direction, as shown in Figure 8. Wind direction angles of 0°–270° and 320°–350° (sectors 1 to 6) were determined to be class D roughness with a ground roughness index $\alpha = 0.30$, while the wind direction angles 280°–310° (sector 7) are class C roughness with $\alpha = 0.22$. The area 2.5 km upstream of the actual incoming wind direction (340°) is mostly densely built up, and the area 2.5–5 km is mostly village farmland, as shown in Figure 9.

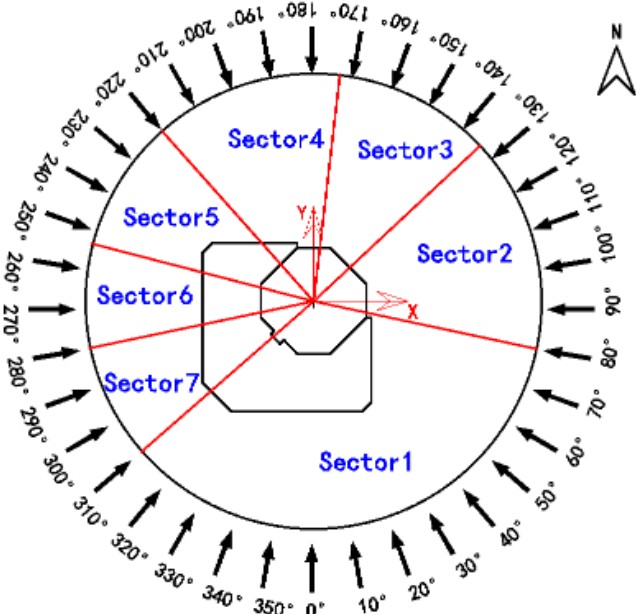

**Figure 7.** Schematic diagram of wind direction angle and structural coordinates.

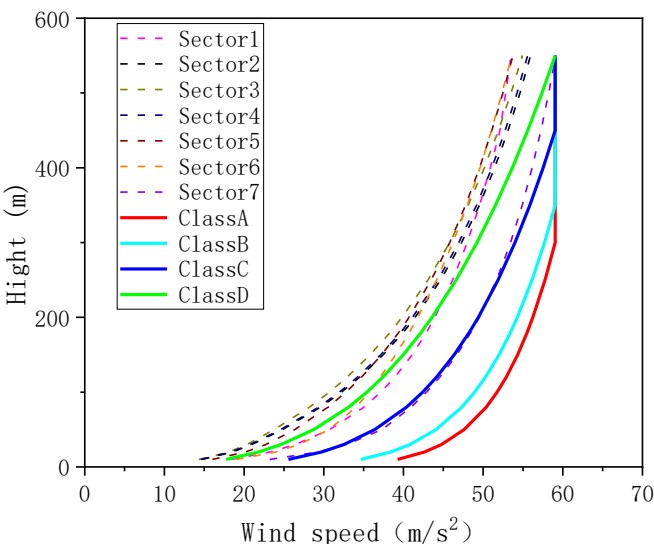

**Figure 8.** Wind profile curve.

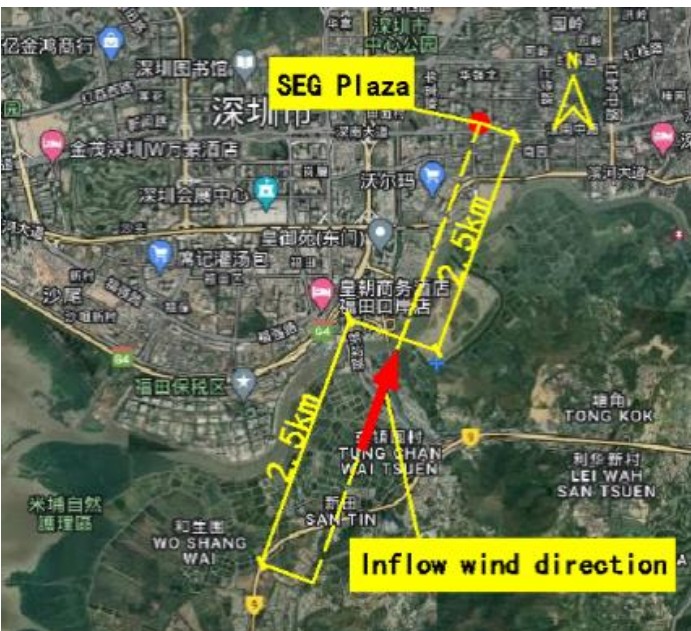

**Figure 9.** Schematic diagram of upstream site.

### 3.2. Calculation Parameters

According to the measurement of [2], the wind speed at the 298 m height of the structure on the day of vibration was 9–12 m/s. The wind speed at the 298 m height was taken as 10.5 m/s for the calculation and analysis in this paper, corresponding to the wind pressure of 0.0186 kPa at the 10 m height for Class C roughness and 0.0090 kPa at the 10 m height for Class D roughness; the wind speeds at the 298 m height for Class C and Class D in the area for the 1-year recurrence period are 28.3 m/s and 25.7 m/s, respectively. The wind pressure values at the 10 m height were 0.112 kPa and 0.054 kPa, respectively. The measured wind speed and wind pressure are lower than the 1-year recurrence period wind speed and wind pressure. The analysis is mainly carried out for the base overturning moment and the peak acceleration at the top. X and Y forward frequencies of the fourth mode and primary torsional frequency were taken for the calculations, as shown in Table 1. As the SEG Plaza has a mixed structure, and the actual wind speed was lower than the 1-year recurrence wind speed in the area, the damping ratio of the structure is 1% for the

1-year return period when calculating the acceleration and 3.5% for the 50-year return period when calculating the base overturning moment according to the *Comfort evaluation standard and control technical specification for wind-induced vibration of tall building* [9], combined with the measured values in the aforementioned literature.

### 3.3. Analysis of Results

Figure 10 shows the variation curve of overturning moment of basement with surrounding buildings with wind angle under wind pressure in 50-year return period. The maximum base overturning moment and the corresponding wind direction and the measured base overturning moment under the 340° wind direction are shown in Table 2. It is evident that the base overturning moment is mainly controlled by the mean wind load downwind, with the most unfavorable wind direction of Mx occurring at 190° and the most unfavorable wind direction of My occurring at 310°. At the 340° wind direction, Mx and My are $2.52 \times 10^9$ N-m and $1.50 \times 10^9$ N-m, respectively. The YJK software is applied to calculate the Mx value of $5.72 \times 10^9$ N-m and My value of $5.73 \times 10^9$ N-m under 50-year recurrence period wind load conditions according to the specification, and the comparison shows that the overturning moments around the *X*-axis and around the *Y*-axis are less than the specification values, which are 44.08% and 26.10% of the specifications, respectively.

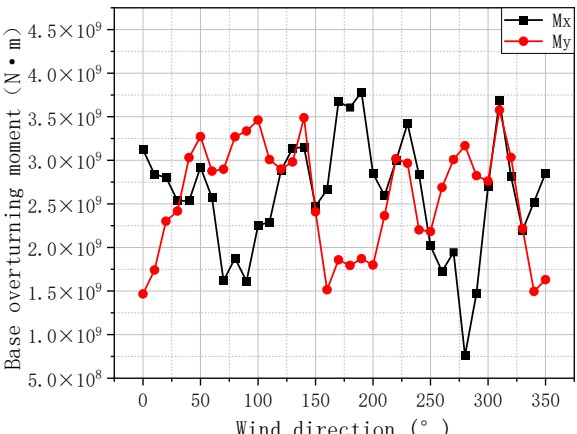

**Figure 10.** Overturning moment curve of base under wind pressure in 50-year return period.

**Table 2.** Overturning moment and wind direction angle.

| Basic Wind Pressure (kN/m²) | Direction | Overturning Moment (N·m) | Wind Direction (°) | Description |
|---|---|---|---|---|
| 0.75 | Mx | $3.78 \times 10^9$ | 190 | Most unfavorable wind direction |
| | My | $3.58 \times 10^9$ | 310 | Most unfavorable wind direction |
| | Mx | $2.52 \times 10^9$ | 340 | Measured wind direction |
| | My | $1.50 \times 10^9$ | 340 | Measured wind direction |

The peak acceleration at the top of the structure indicates the effect of fluctuating wind loads on the wind-induced vibration response of the structure and is one of the major indicators for comfort evaluation. The centroid and angular acceleration response of the highest used floor of the structure (288.11 m) was extracted for evaluation, with the extraction position shown in Figure 11. The centroid acceleration curve of the structure at 10.5 m/s wind speed are shown in Figure 12, and the acceleration of the centroid at the most unfavorable wind direction and wind direction of 340° are presented in Table 3. The maximum acceleration can be seen in wind direction of 310°, and the acceleration in the wind direction of 340° with a perimeter building is smaller as compared to other wind directions, reaching 70.50 milligal in X-direction and 84.76 milligal in Y-direction, which is 10.61% higher than the acceleration in Y-direction without a perimeter working in this wind direction. The angular acceleration at the wind direction of 340° wind angle

is shown in Table 4. It can be seen that there is a maximum angular acceleration of 86.21 milligal at angular 7 and 8 of the building with perimeter, which is 10.61% higher than that of the condition without perimeter. This finding is consistent with the influence law of the interference effect in [10,11]. According to the comfort evaluation criteria, the acceleration limit corresponding to 0.15 Hz at the first-order frequency of the structure is $4.6 \times 103$ milligal, and the acceleration response value is much smaller than the limit value, meeting the comfort requirements.

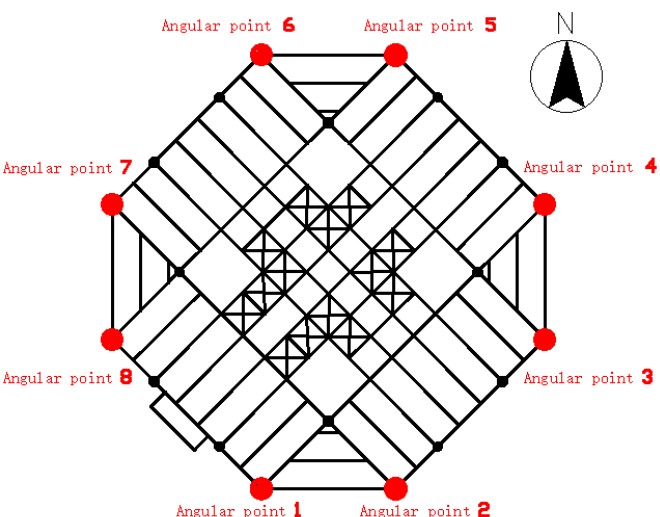

**Figure 11.** Schematic diagram of angular acceleration extraction position.

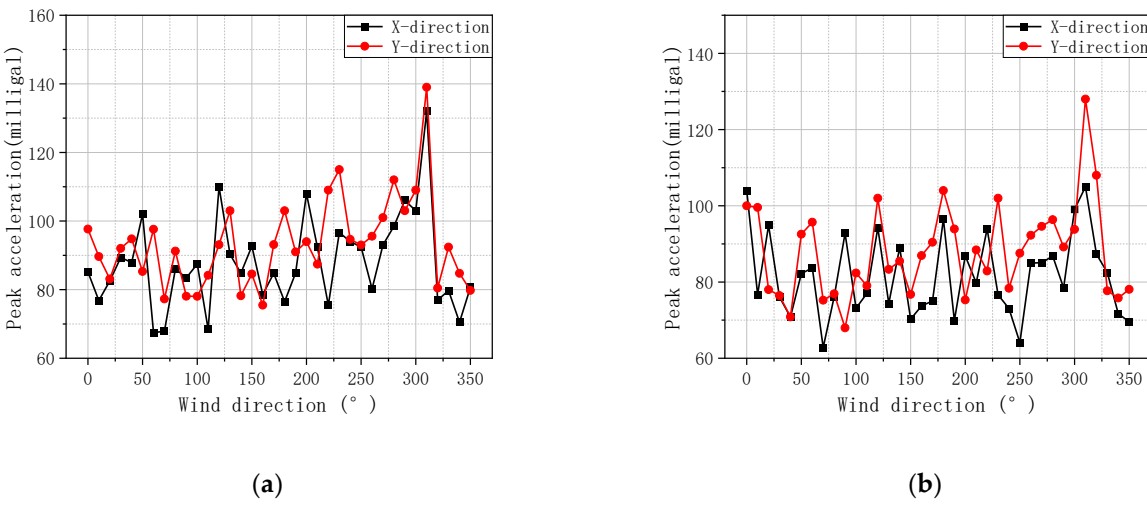

(**a**)                                           (**b**)

**Figure 12.** Top centroid acceleration curve. (**a**) With perimeter buildings; (**b**) No perimeter building.

**Table 3.** Acceleration and wind direction angle.

| Operating Conditions | Direction | Acceleration (Milligal) | Wind Direction (°) | Description |
|---|---|---|---|---|
| With perimeter buildings | X | 132.0 | 310 | Most unfavorable wind direction |
| | Y | 139.0 | 310 | Most unfavorable wind direction |
| | X | 70.5 | 340 | Measured wind direction |
| | Y | 84.8 | 340 | Measured wind direction |
| No perimeter building | X | 105.0 | 310 | Most unfavorable wind direction |
| | Y | 128.0 | 310 | Most unfavorable wind direction |
| | X | 71.7 | 340 | Measured wind direction |
| | Y | 75.8 | 340 | Measured wind direction |

**Table 4.** Angular acceleration.

| Operating Conditions | Angular Acceleration (Milligal) | | | | | | | |
|---|---|---|---|---|---|---|---|---|
| | 1 | 2 | 3 | 4 | 5 | 6 | 7 | 8 |
| With perimeter buildings | 85.83 | 85.82 | 86.16 | 86.16 | 85.81 | 85.83 | 86.21 | 86.21 |
| No perimeter building | 76.11 | 76.09 | 76.34 | 76.35 | 76.10 | 76.10 | 76.39 | 76.40 |

## 4. Sensitivity Analysis

The acceleration of the main structure at a wind speed of 10.5 m/s is much smaller than the 1-year wind vibration comfort limit. As the acceleration response of the main structure is related to the wind speed, the self-vibration frequency of the structure and the damping ratio, and given that the wind speed is uncertain, the self-vibration frequency of the structure is affected by construction and deviates from the design value and there is no definite value for the damping ratio, it is necessary to analyze the acceleration response of the structure at different wind speeds, frequencies and damping ratios.

### 4.1. Calculation Parameters

As mentioned earlier, the wind speed at the 298 m height measured on the day of vibration was 9–12 m/s, so the wind speeds of 9 m/s and 12 m/s were taken for analysis. 9 m/s corresponds to wind pressures of 0.0137 kPa and 0.0066 kPa at 10 m height for class C and D roughness, respectively, and 12 m/s equates to wind pressures of 0.0244 kPa and 0.0117 kPa at 10 m height for class C and D roughness, respectively. The tower building makes up the major part of the main structure and the frequency variation of the completed tower building ranges from −5% to +20%, and ±5% frequency is taken for the analysis and the frequency taken is shown in Table 5, where the measured frequency is 2.12 Hz. Combined with the previously measured damping ratios of 1.08% and 0.35%, respectively, the damping ratios of 1.08% and 0.35% are taken for the sensitivity analysis of the main structure. Notably, the sensitivity analysis is performed based on the wind load test data with the surrounding buildings.

**Table 5.** Value of vibration frequency of main structure for sensitivity analysis.

| Frequency Change | Participating Mass Ratio | 1 Mode | 2 Mode | 3 Mode | 4 Mode | 1 Mode/Measured (%) |
|---|---|---|---|---|---|---|
| | UX | 0.1632 | 0.6779 | 1.3950 | 1.5178 | 7.7 |
| 5% | UY | 0.1593 | 0.6122 | 1.2819 | 1.8505 | 7.5 |
| | UZ | 0.3136 | | | | 14.8 |
| | UX | 0.1477 | 0.6134 | 1.2621 | 1.3732 | 7.0 |
| −5% | UY | 0.1442 | 0.5539 | 1.1598 | 1.6743 | 6.8 |
| | UZ | 0.2837 | | | | 13.4 |

### 4.2. Analysis of the Results

The centroid acceleration at the top under the wind direction of 340° is shown in Figure 13. The corner acceleration at the top under the wind direction of 340° is shown in Figure 14. The acceleration varies significantly for different wind speeds and damping ratio conditions, with the acceleration increasing with wind speed and decreasing with damping ratio, which is consistent with the effect of damping ratio on the acceleration of the centroid obtained in [12]. At a damping ratio of 0.35%, the acceleration varies significantly with frequency, and at damping ratios of 1% and 1.08%, the acceleration varies insignificantly with frequency. The wind speed is 12 m/s, the frequency is +5% and the damping ratio is 0.35%, which is the most unfavorable operating condition for the X- and Y-directional centroid acceleration, where the peak acceleration in the X-direction is 235 milligal and the peak acceleration in the Y-direction is 216 milligal.

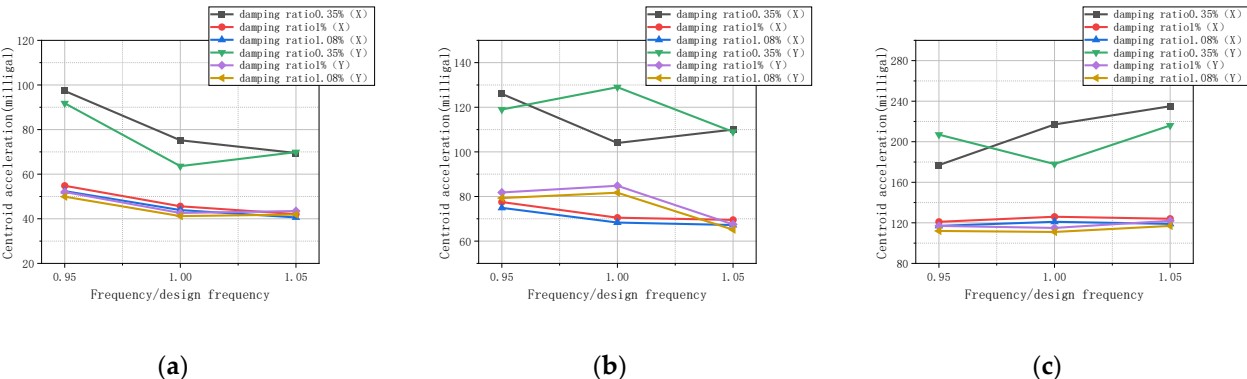

(**a**)　　　　　　　　　　　　　(**b**)　　　　　　　　　　　　　(**c**)

**Figure 13.** The 340-degree centroid acceleration curve based on sensitivity analysis. (**a**) 340° 9 m/s; (**b**) 340° 10.5 m/s; (**c**) 340° 12 m/s.

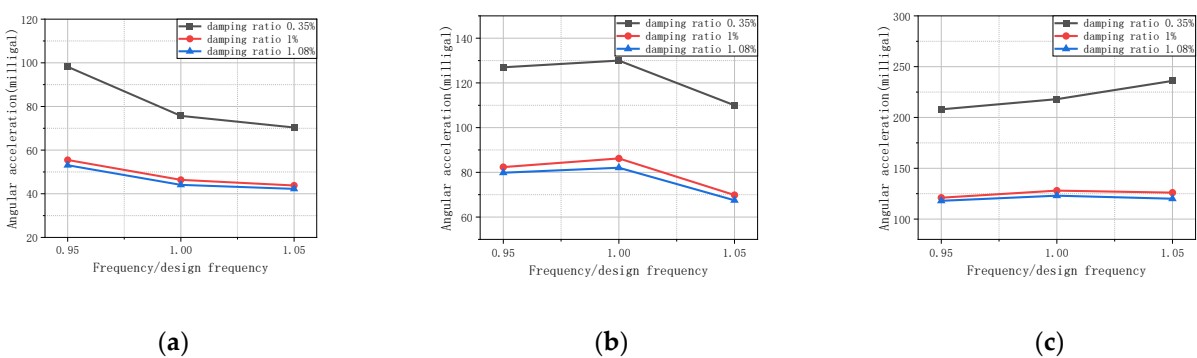

(**a**)　　　　　　　　　　　　　(**b**)　　　　　　　　　　　　　(**c**)

**Figure 14.** The 340-degree angular acceleration curve based on sensitivity analysis. (**a**) 9 m/s; (**b**) 10.5 m/s; (**c**) 12 m/s.

Given the deviation of wind direction, the acceleration analysis was carried out at the wind directions of 330° and 350°, respectively, with the same wind speed of 12 m/s, frequency of +5% and damping ratio of 0.35%, which is the most unfavorable working condition for X-direction and Y-direction centroid acceleration. The peak X-directional acceleration at a wind direction of 350° is 215 milligal and the peak Y-directional acceleration is 200 milligal.

From the aforementioned analysis, it can be seen that the measured wind speed on the day of vibration corresponds to a small wind pressure. Therefore, the acceleration limits for the 1-year recurrence period of the *Comfort evaluation standard and control technical specification for wind-induced vibration of tall buildings* (DBJ/T 15-216-2021) [8] for buildings in Guangdong are adopted to evaluate the 340° wind-directional angular acceleration, as shown in Figure 15. By referring to the Japanese AIJ comfort evaluation standard [13], the specification establishes four evaluation curves that allow the comfort evaluation of structures with different self-vibration frequencies, where H-90, H-70, H-50 and H-30 indicate that there are 90%, 70%, 50% and 30% personnel perception rates, respectively. As can be observed from the above results, at a wind speed of 12 m/s, a 5% increase in frequency and a damping ratio of 0.35%, both the centroid acceleration and the angular acceleration reach a maximum of 235 milligal in the X-direction, 216 milligal in the Y-direction and 236 milligal in the angular at a wind direction of 340°, which all correspond to the first mode frequency of the structure of 0.1632 Hz. The accelerations are evaluated according to the 1-year return period acceleration limits, which are all much less than the required limits, and the accelerations at the wind directions of 330° and 350° are also much less than the required limits. Therefore, it can be concluded that the impact of wind loads on the main structure of the SEG Plaza is relatively small and was not the dominant cause of the multiple vibrations from May 18 to 20 that year.

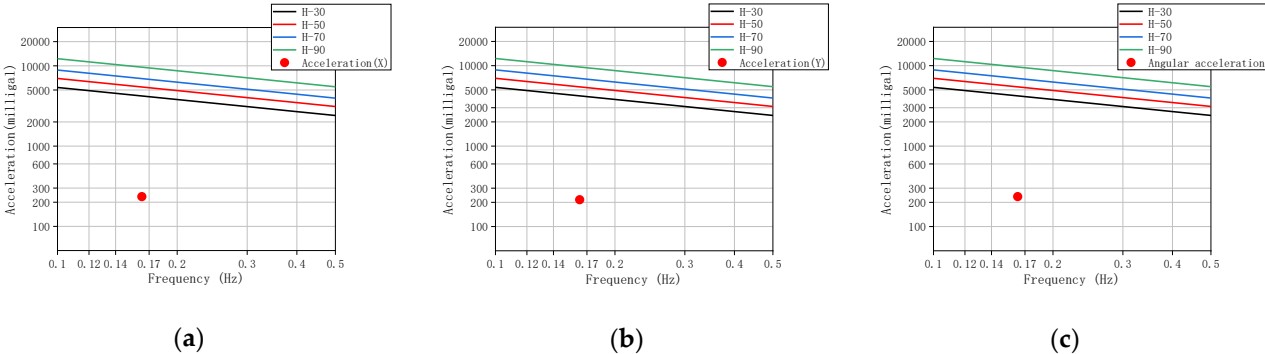

**Figure 15.** Comfort evaluation curve. (**a**) X-direction; (**b**) Y-direction; (**c**) Angular.

## 5. Response Analysis of the Main Structure with the Factor of Mast Included

According to the previous analysis, the comfort level of the main structure is satisfactory when the mast at the top is not considered, and the wind load on the main structure is not the major cause of vibration. In order to further investigate the causes of unusual vibration of the building, it is necessary to analyze the structural vibration response by considering the mast.

### 5.1. Load Calculation on Vortex-Induced Resonance of the Mast

Under certain conditions, the cylinder under the action of the wind undergoes Kármán vortex street phenomenon [14,15], and even in the high Reynolds regions, there was still periodic vortex shedding in the cylinder wake [16–19], and lift perpendicular to the incoming flow was generated [20]. If the vortex shedding frequency is close to the high frequency of the structure, the locking phenomenon occurs [21–23], which easily leads to vortex-induced resonance.

The structure of the top mast is long and slender, with relatively low damping and is susceptible to vortex-induced vibration. The lower half is a lattice structure with high stiffness, and the upper half is a cantilever section with high flexibility and a cylindrical structure, which is the key area of interest for the analysis of vortex-induced resonance.

Based on the measured vibration frequency of 2.12 Hz and the results of the modal analysis of the mast using ETABS software, the third-mode self-vibration frequency of the mast is close to the measured frequency of 2.14 Hz and the direction of vibration of the third-mode vibration pattern was close to perpendicular to the measured wind direction of 340°. As the vortex-induced vibration is in the cross-wind direction, the direction of vibration is perpendicular to the direction of the incoming wind, so it is assumed that the mast is experiencing vortex-induced vibration of the third-mode vibration pattern.

Firstly, the equivalent wind load is determined for the cross-wind vibration of a circular section structure. The expression for the critical wind speed of the vortex-induced resonance is:

$$\overline{v}_{cr} = \frac{D}{T_j S_t} = \frac{n_j D}{S_t} \tag{1}$$

where $T_j$ and $n_j$ are the $j$th mode self-vibration period and self-oscillation frequency of the structure, $D$ is the cylindrical diameter, $S_t$ is the Strouhal number, the $S_t$ of the cylinder is taken as 0.2. Taking the measured vibration frequency of 2.12 Hz, the cylindrical diameter is taken as 1.1 m of the cantilever section, and the critical wind speed is calculated as 11.66 m/s.

The structure will resonate within a certain wind speed range. For the cylindrical structure, a height region of $1$–$1.3\overline{v}_{cr}$ is taken as the resonance zone along the height

direction. The expressions for the starting height of the resonance zone, $H_1$, and the top height of the resonance zone, $H_2$, are:

$$H_1 = 10 \left( \frac{\overline{v}_{cr}}{\overline{v}_{oa}} \right)^{\frac{1}{\alpha}} \tag{2}$$

$$H_2 = 10 \left( \frac{1.3\overline{v}_{cr}}{\overline{v}_{oa}} \right)^{\frac{1}{\alpha}} \tag{3}$$

where $\overline{v}_{oa}$ is the actual wind speed at a height of 10 m for this ground roughness category, ff is the ground roughness index, and the actual wind direction corresponds to roughness category $D$ and ff is taken as 0.3. The mast is not in the resonance zone at 298 m wind speeds of 9 m/s and 10.5 m/s, and no third-mode vortex vibration occurs. At a wind speed of 12 m/s at 298 m, $H_1$ is 278 m and $H_2$ is 666.7 m, all parts of the mast are within the resonance zone and the mast experiences three-mode vortex-induced vibration at this wind speed.

The equivalent wind load amplitude $P_{dji}$ at resonance is calculated using the layer concentrated force as blows:

$$P_{dji} = \lambda_j \frac{\varphi_{ji}\overline{v}_{cr}{}^2 D_0 h_i}{12800\xi_j} \tag{4}$$

where $h_i$ is the height of the $i$th level, $\varphi_{ji}$ is the j mode coefficient, $D_0$ is the cylindrical diameter, $\xi_j$ is the damping ratio of $j$ mode, $\lambda_j$ can be presented as:

$$\lambda_j = \frac{\int_{H_1}^{H} \varphi_j(z) dz}{\int_0^H \varphi_j{}^2(z) dz} \tag{5}$$

The modal analysis of the mast is conducted using ETABS to obtain the vibration coefficient of the third mode, and $\lambda_3$ of 1.53 was obtained. The cumulative damage caused by the long-term vibration of the mast is considered to reduce the damping ratio and according to [24] and the damping ratio of the wind turbine tower is obtained to be in the range of 0.1–0.3% if the cumulative damage is included. The mast and the wind turbine tower are both round steel tubes, and the damping ratio $\xi_3$ of the mast is taken as 1% and 0.3%, respectively. $P_{d3i}$ and $P_0$, the equivalent wind load, are 42.96 kN and 143.19 kN for each layer when substituted into Equation (4). $P_0$ is the equivalent counter-force of the mast acting on the tower.

Since the vortex shedding produces cross-wind effect on the surface of the structure, and the third-mode vibration direction of the mast is shown in Figure 5. The direction of the counter-force at the bottom of the mast is the 45° clockwise rotation direction of the $X$-axis, projecting $P_0$ to the X- and Y-axes as the wind load component, according to the mast offset from the center of the tower by 20.25 m to obtain the moment, and the time range of the counter-force at the bottom of the mast is considered as a simple harmonic load.

$$V(t) = P_0 \sin(2\pi \times n_j t) \tag{6}$$

This leads to a mast reaction time of $V(t) = P_0 \sin(4.24\pi t)$.

See Supplementary Materials for specific calculation process and data.

### 5.2. Acceleration Response of the Main Structure with the Factor of Mast Included

It is found that no vortex resonance occurred in the mast at 9 m/s and 10.5 m/s, so in this section only the acceleration analysis is carried out for the 12 m/s condition. In order to obtain the acceleration response of the main structure under vortex-induced vibration of the mast, the time course counter-force at the bottom of the mast is superimposed with the time course of the wind load at the top of the main structure and the acceleration response of the main structure is calculated by inputting the first 20 mode of vibration.

Considering the acceleration response of the center of mass under the 340 wind direction angle of the mast, as shown in Figure 16. When the mast damping ratio is taken as 1%, the acceleration increases with the structure frequency and decreases with the increase of the damping ratio. The maximum accelerations in both X and Y directions are obtained when the structure frequency is increased by 5% and the structural damping ratio is 0.35%. At a mast damping ratio of 0.3%, the X-direction acceleration is 1962 milligal and the Y-direction acceleration is 956 milligal.

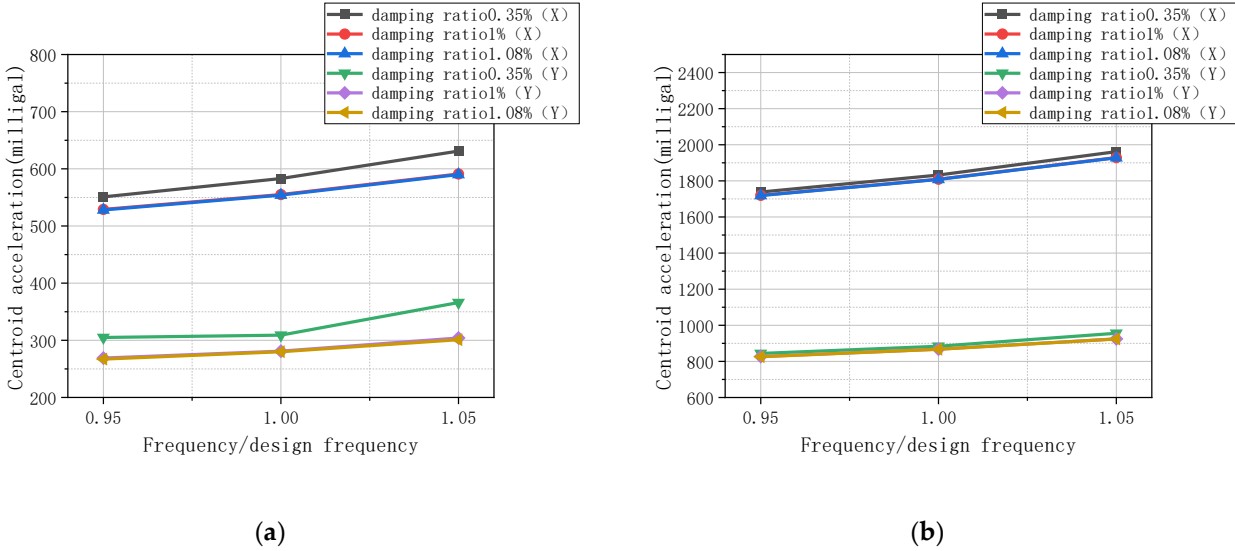

(**a**)         (**b**)

**Figure 16.** Considering the centroid acceleration of mast. (**a**) At the mast damping ratio of 1%; (**b**) At the mast damping ratio of 3%.

The maximum centroid acceleration is obtained at the same frequency and damping ratio for wind directions of 330° and 350°, with a mast damping ratio of 1% and 617 milligal in the X-direction and 366 milligal in the Y-direction for the 330° wind angle and 620 milligal in the X-direction and 335 milligal in the Y-direction for the wind direction of 350°. At a mast damping ratio of 0.3%, the X-direction acceleration is 1960 milligal and the Y-direction acceleration is 949 milligal for the wind direction of 330°, and the X-direction acceleration is 1956 milligal and the Y-direction acceleration is 944 milligal for the wind direction of 350°.

A comparison of the results for the two mast damping ratios reveals that the value of the mast damping ratio has a significant effect on the centroid acceleration of the structure, mainly because the mast damping ratio is inversely proportional to its equivalent wind load amplitude, and therefore the decrease in damping ratio due to cumulative damage to the top mast cannot be ignored. Compared with frequency, the effect of the structural damping ratio on the acceleration of the center of mass is more obvious.

The angular acceleration response of the mast at the wind direction of 340° is shown in Figure 17. When the mast damping ratio is set at 1%, the maximum angular acceleration is 2615 milligal for a 5% increase in structural frequency and a structural damping ratio of 0.35%, while the maximum angular acceleration is 8687 milligal for a mast damping ratio of 0.3%. The mast damping ratio also significantly affects the acceleration of the structure. Different from centroid acceleration, the most important factor affecting angular acceleration is the structural frequency.

The maximum angular acceleration is obtained at the same frequency and structural damping ratio for wind directions of 330° and 350°, with a mast damping ratio of 1% giving 2612 milligal at the wind direction of 330° and 2613 milligal at the wind direction of 350°. At a mast damping ratio of 0.3%, the acceleration at the wind direction of 330° is 8686 milligal and 8685 milligal at the wind direction of 350°.

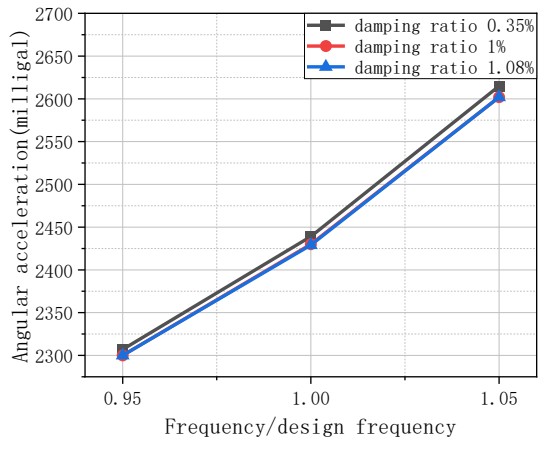
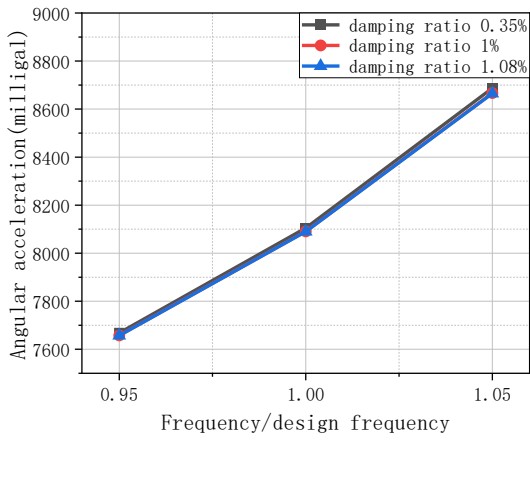

| (**a**) | (**b**) |

**Figure 17.** Considering the angular acceleration of mast. (**a**) At the mast damping ratio of 1%; (**b**) At the mast damping ratio of 3%.

The acceleration response of the mast under the most unfavorable operating condition of the wind direction of 340° is evaluated, as shown in Figure 18. When the mast damping ratio is taken as 1%, the maximum acceleration in the X direction is 631 milligal and the maximum acceleration in the Y direction is 366 milligal, both of which are lower than the H-30 curve. The maximum acceleration at the angular is 2615 milligal, which is above the H-70 curve, and when the acceleration perception rate of people is above 70%, the comfort level is poor. At 0.3% mast damping ratio, the maximum acceleration in the X direction is 1962 milligal, which is above the H-50 curve. The maximum acceleration in the Y direction is 956 milligal, which is lower than the H-30 curve, and the maximum angular acceleration is 8687 milligal, which is much higher than the H-90 curve, and when the acceleration perception rate of people is above 90%, the comfort level is poor. The excessive acceleration levels at winds direction of 330° and 350° wind angles is consistent with that at the wind direction of 340°.

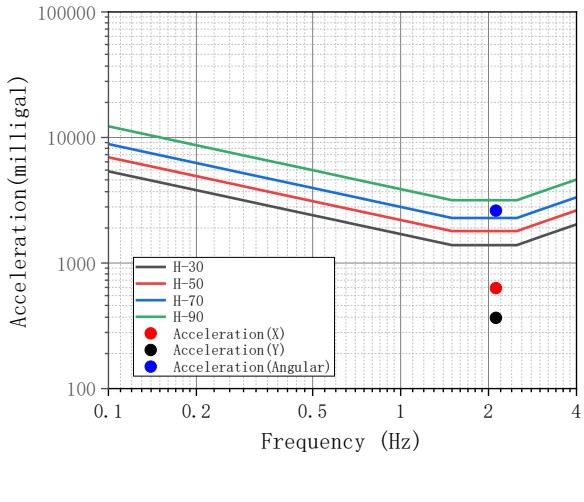
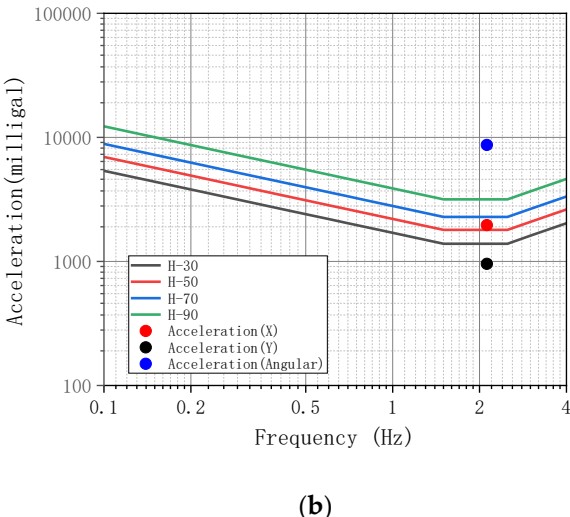

| (**a**) | (**b**) |

**Figure 18.** Comfort evaluation considering mast. (**a**) At the mast damping ratio of 1%; (**b**) At the mast damping ratio of 0.3%.

The main reason for this is that the three-mode vortex-induced resonance of the mast, which excites the main structure, causes a significant increase in acceleration, and the

acceleration threshold for human perception of vibration at the resonance frequency of 2.12 Hz is lower than the acceleration threshold for human perception of vibration at the first mode self-vibration frequency of the main structure of 0.1632 Hz. This provides a greater degree of validation that the major cause of vibration in the SEG Plaza is the third-mode vortex-induced resonance of the top mast, which triggers the higher resonance of the main structure.

## 6. Conclusions

In this paper, a study was conducted on the unusual vibration of a structure in the SEG Plaza. A rigid model wind tunnel test was carried out on the main structure without the factor of the mast included to analyze the base overturning moment and the acceleration at the top of the structure.

1.  Under the 50-year return period wind pressure condition, the base overturning moment of the main structure excluded the mast factor in the X-direction and Y-direction, which are less than the specification values of 44.08% and 26.10% of the specification ones, respectively; under the measured wind speed and direction condition (10.5 m/s and at the wind direction of 340°), the acceleration was much smaller than the acceleration limit value of the standard, which meets the comfort evaluation requirement.
2.  Sensitivity analysis of the acceleration of the main structure with the mast factor excluded reveals that the acceleration increases with the increase of wind speed and decreases with the increase of damping ratio, the acceleration at the centroid acceleration and the angular acceleration under the most unfavorable operating conditions are lower than the H-30 curve. As a result, it can be concluded that the wind load on the main structure of SEG Plaza had little effect on the vibration of the main structure and was not the major cause of the vibration.
3.  The mast resonates at a 12 m/s wind speed, and the acceleration response of the main structure with mast exceeds the limit when the mast damping ratio is 1% and 0.3%, respectively, in which the angular acceleration even exceeds the limit of H-90 when the mast damping ratio is 0.3%. This indicates that the vibration of the SEG Plaza was mainly due to the third-mode vortex-induced resonance of the top mast, which triggers higher resonance of the main structure.

**Supplementary Materials:** The following supporting information can be downloaded at: https://www.mdpi.com/article/10.3390/su15043067/s1.

**Author Contributions:** Conceptualization, W.X., R.L. and J.Q.; methodology, W.X., R.L. and J.Q.; software, R.L.; validation, W.X., R.L. and J.Q.; formal analysis, R.L.; investigation, J.Q.; resources, W.X. and Q.L.; data curation, R.L. and J.Q.; writing—original draft preparation, R.L.; writing—review and editing, W.X. and J.Q.; visualization, W.X. and Z.Y.; supervision, Q.L. and Z.Y.; project administration, W.X.; funding acquisition, W.X. All authors have read and agreed to the published version of the manuscript.

**Funding:** This research was funded by [Guangdong Provincial Building Research Institute Group Co., Ltd.] grant number [0100RDY2021F0000386] and The APC was funded by [Guangdong Provincial Building Research Institute Group Co., Ltd.].

**Institutional Review Board Statement:** Not applicable.

**Informed Consent Statement:** Not applicable.

**Data Availability Statement:** Data is contained within the article or Supplementary Material.

**Conflicts of Interest:** The authors declare no conflict of interest.

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
