# Peer review of "Study on Wind-Induced Human Comfort of the SEG Plaza under Local Excitation Based on Wind Tunnel Test"

_sustainability, doi:10.3390/su15043067_

Round 1

Reviewer 1 Report

The current manuscript provides a study on wind-induced human comfort of the SEG Plaza under local excitation based on wind tunnel test. It is interesting, but I have several comments that need to be addressed before a final decision can be made on the paper:

1. I am concerned about the total number of references. The total number of references for the regular paper should have a minimum of 25 papers.

2. Fig. 6, for the rigid scale model, I wonder how the authors considered the flow around the SEG Plaza since another building may block the incoming flow.

3. Fig. 7 shows the wind speed for sector 7 is the highest of the others. Did the authors already consider the phenomena? 

4. How the authors calculate the data is not so obvious. I suggest adding more explanation to the calculation parameter section.

5. Figure 15, it is interesting that the comfort evaluation curve for X, Y, and Angular direction shows a similar trend even. The authors suggest explaining the reason behind the behavior as well as which factor affects mostly for the comfort evaluation.

6. Section 4.1, What is the consideration of taking value for the calculation? Since I believed that all the calculations were strongly affected by the properties and direction of the incoming wind.  

Author Response

Thank you for your suggestions. I will reply to your questions one by one in the word file.

Reviewer 2 Report

Paper title: Study on wind-induced human comfort of the SEG Plaza under 1 local excitation based on wind tunnel test

Manuscript ID: sustainability-2151932

General comments: This paper introduce very interesting topic in which try to solve real case problem occurred in SEG Plaza from May 18 to 20, 2021. The study used the wind- induced response of the main structure to simulate the vibration, and acceleration sensitivity analysis is conducted with parameters such as wind speed, structural period and damping ratio included.

Comments to authors:

1. In abstract “the mast as per the current specifications, ”; “This indicates that the wind load on the main structure is not the dominant cause of the structural vibration.”is not clear

2. Pls., follow the journal format as the numbering in introduction should start by 1.

3. Is better to introduce different subsection two and three subtitle, (Calculation parameters, and Analysis of the results)

4. Redraw figure 15 and figure 16.

5. In addition, is need to introduce the tool for measurement, you can check these papers

https://link.springer.com/article/10.1007/s11356-022-19902-8

Author Response

(The authors gave the same response as above.)

Round 2

Reviewer 1 Report

Thank you for the revised version of your manuscript. However, it needs a small modification in Section 4.1., I still could not find the details calculation used in the current study. Please provide an example for the data step calculation.

Author Response

Thank you for your question. See Annex 1 for the detailed calculation process. Because there are many calculation processes, in order to simplify the article, I don't intend to add the specific calculation process to it, but only show the calculation method in Section 4.1.
